# Gut-Microbiota-Derived Metabolites Maintain Gut and Systemic Immune Homeostasis

**DOI:** 10.3390/cells12050793

**Published:** 2023-03-02

**Authors:** Juanjuan Wang, Ningning Zhu, Xiaomin Su, Yunhuan Gao, Rongcun Yang

**Affiliations:** 1Department of Immunology, Nankai University School of Medicine, Nankai University, Tianjin 300071, China; 2Translational Medicine Institute, Affiliated Tianjin Union Medical Center of Nankai University, Nankai University, Tianjin 300071, China; 3State Key Laboratory of Medicinal Chemical Biology, Nankai University, Tianjin 300071, China

**Keywords:** gut microbiota, SCFAs, tryptophan metabolites, bile acid metabolites, tolerogenic macrophages, regulatory T cells

## Abstract

The gut microbiota, including bacteria, archaea, fungi, viruses and phages, inhabits the gastrointestinal tract. This commensal microbiota can contribute to the regulation of host immune response and homeostasis. Alterations of the gut microbiota have been found in many immune-related diseases. The metabolites generated by specific microorganisms in the gut microbiota, such as short-chain fatty acids (SCFAs), tryptophan (Trp) and bile acid (BA) metabolites, not only affect genetic and epigenetic regulation but also impact metabolism in the immune cells, including immunosuppressive and inflammatory cells. The immunosuppressive cells (such as tolerogenic macrophages (tMacs), tolerogenic dendritic cells (tDCs), myeloid-derived suppressive cells (MDSCs), regulatory T cells (Tregs), regulatory B cells (Breg) and innate lymphocytes (ILCs)) and inflammatory cells (such as inflammatory Macs (iMacs), DCs, CD4 T helper (Th)1, CD4Th2, Th17, natural killer (NK) T cells, NK cells and neutrophils) can express different receptors for SCFAs, Trp and BA metabolites from different microorganisms. Activation of these receptors not only promotes the differentiation and function of immunosuppressive cells but also inhibits inflammatory cells, causing the reprogramming of the local and systemic immune system to maintain the homeostasis of the individuals. We here will summarize the recent advances in understanding the metabolism of SCFAs, Trp and BA in the gut microbiota and the effects of SCFAs, Trp and BA metabolites on gut and systemic immune homeostasis, especially on the differentiation and functions of the immune cells.

## 1. Introduction

The gut microbiota is established at birth and evolves with age, and also maintains a commensal relationship with the host, being an integral part of the human body. The mammalian gastrointestinal tract harbors large amounts of different gut microbiota [1], including bacteria, archaea, fungi, viruses and phages. These gut microorganisms not only participate in food and energy metabolism but also contribute to the host immune response and homeostasis [2,3]. The alteration of the gut microbiota can lead to the occurrence and development of many diseases [4].

In recent years, with the rapid development of molecular biology, genomics, bioinformatics analyses and high-throughput sequencing techniques, great progress has been made in understanding the gut microbiota with diseases [5] such as neurodegenerative diseases (Parkinson’s disease and Alzheimer’s disease), cardiovascular diseases (hypertension and atherosclerosis), metabolic diseases (obesity, diabetes, and non-alcoholic fatty liver disease (NAFLD)), and gastrointestinal diseases (inflammatory bowel diseases (IBD) and colorectal cancer (CRC)). These effects on the health of the host can occur through many ways such as energy absorption [5] and the microbiota–gut–brain axis [6]. However, the roles of altered gut microbiota in diseases are related to gut microbiota metabolites such as short-chain fatty acids (SCFAs), tryptophan (Trp) and bile acid (BA) metabolites from different microorganisms. The effects of gut microbiota metabolites on the local and systemic immunity have already attracted much attention. A growing body of clinical evidence has suggested an intricate relationship between the gut microbiota and the immune system. Most altered-gut-microbiota-mediated diseases are related to impaired immune responses [7].

Gut-microbiota-derived metabolites not only affect genetic and epigenetic regulation but also impact the metabolism of the immune cells via their receptors in the immune cells [8,9,10]. These metabolites from different microorganisms can not only promote the differentiation and function of immunosuppressive cells but also inhibit the inflammatory cells, together maintaining the gut and systemic immune homeostasis of the individuals (Figure 1). Since there are three main specific classes of metabolites, namely SCFAs, Trp and BA metabolites, that have been found in the gut microbiota so far, we here will summarize the recent advances in understanding the metabolism of SCFAs, Trp and BA in different microorganisms and the effects of SCFAs, Trp and BA metabolites on the gut and systemic immune homeostasis, especially on the differentiation and functions of immune cells.

## 2. Gut Microbiota and Metabolites

### 2.1. Gut Microbiota and Short-Chain Fatty Acids

Short-chain fatty acids (SCFAs) are carboxylic acids produced from dietary fiber fermentation in the cecum and colon by gut bacteria [11,12] (Table 1), mainly including acetate (C2), propionate (C3) and butyrate (C4).

### 2.2. Gut Microbiota and Tryptophan Metabolites

Tryptophan (Trp) metabolism in the gut microbiota has been reviewed by us [18] and others [19,20,21]. Trp can be converted into various metabolites by the gut microbiota (Table 2) such as indole, indole-3-aldehyde (IAld), indole-3-acid-acetic (IAA), tryptamine, indoleacrylic acid (IA), indole ethanol (IE), indole-3-propionic acid (IPA), indole-3-acetaldehyde (IAAld) and 3-methylindole (skatole). Trp also produces kynurenine (Kyn) and downstream metabolites such as 3-hydroxykynurenine (3H-Kyn) and 3-hydroxyanthranilic acid (3-HAA) [19,20,21].

### 2.3. Gut Microbiota and Bile Acid Metabolites

Two primary bile acids (BAs), i.e., cholic acid (CA) chenodeoxycholic acid (CDCA) are generated in the liver. These primary BAs can be conjugated, deconjugated and transformed into other metabolites in the gut microbiota (Table 3). Primary BAs are conjugated with glycine, taurine or other amino acids in hepatocytes and also in the gut microbiota. Conjugated BAs derived from the liver can be deconjugated in the gut microbiota through bile salt hydrolases (BSHs) in the small intestine [30]. While BAs are deconjugated, BAs can be converted into secondary BAs, i.e., DCA and lithocholic acid (LCA). There are four distinct ways to transform BAs, including deconjugation, dehydroxylation, oxidation and epimerization in human [31]. A range of oxo-, epi- and iso-derivatives by microbes [32] is found, such as the oxo-bile acid metabolites 3-oxoLCA, 7-oxoCA, 7-oxoCDCA, 12-oxoCA and 12-oxoDCA [33], and others such as iso-LCA, 3-oxo-LCA, allo-LCA, 3-oxoallo-LCA, isoalloLCA, 3-ketoLCA, LCA acetate and LCA propionate [31,34,35]. Chenodeoxycholic acid (CDCA) can be converted to UDCA [36] and DCA to iso-DCA by 7α-hydroxysteroid dehydrogenase (7α-HSDH) and 7β-HSDH dehydrogenate [37]. The metabolites of DCA, a 3β-hydroxydeoxycholic acid (isoDCA) has been also identified [38]. However, more gut bacterium species that produce BA metabolites still need to be identified.

## 3. Receptors of Gut-Microbiota-Derived Metabolites in the Immune Cells

### 3.1. Receptors of Short-Chain Fatty Acids

Several different ways such as passive diffusion, transporters and receptors help SCFAs enter cells. SCFA absorption can be promoted by the proton-coupled monocarboxylate-transporter 1 (MCT1) and the sodium-coupled monocarboxylate-transporter 1 (SMCT1) promote. Free-fatty acid receptor (FFAR) 2, G-protein coupled receptor (GPR) 43, FFAR3 (GPR41), hydroxycarboxylic acid receptor 2 (HCAR2) (also called GPR109a), Olfr-78 (OR51E2) in humans and Olfr-87 in mice can be activated by SCFAs [53,54]. The SCFAs acetic, propionic and butyric acids mainly activate GPR43 and/or GPR41, whereas butyric and β-hydroxybutyric acids are stimulators of GPR109a. In addition, SCFAs, mainly propionic and butyric acids, also participate in the activation of the peroxisome proliferator-activated receptor γ (PPARγ) [55].

### 3.2. Receptors of Tryptophan Metabolites

The aryl hydrocarbon receptor (AhR) can be activated by various endogenous and exogenous polycyclic aromatic hydrocarbon ligands such as Trp metabolites [56]. This AhR can sense a wide range of intestinal signals, maintaining homeostasis between the gut microbiota and host [57,58]. After activation, a complex of inactive AhRs located in the cytoplasm with the AhR nuclear translocator protein (ARNT) is formed and translocated to the nucleus to control transcriptional activity. Notably, AhR interactions with other proteins are only triggered by specific AhR ligands. This indicates that the specific protein complexes may be induced by different AhR ligands. For AhR activation, indole, skatole, IA, tryptamine, IPyA and indole-3-acetamide (IAM) are the most effective, but IAA, IAID, IPA and ILA are less active [59,60]. Additionally, pregnant X receptor (PXR) can also be recognized by Trp metabolites [61]. The Trp metabolite indole and its derivatives through AhR and PXR contribute to anti-inflammatory activities [62].

### 3.3. Receptors of Bile Acid Metabolites

The receptors of BAs and their metabolites include nuclear and membrane receptors, which have been reviewed by Biagioli et al. [63]. These receptors include nuclear receptors such as farnesoid X receptor (FXR), liver-X-receptor (LXR), vitamin D receptor (VDR), PXR, retinoid related orphan receptor (RORγt), constitutive androstane receptor (CAR) and cell membrane receptors such as G-protein BA receptor 1 (GPBAR1) (or Takeda G protein-coupled receptor 5 (TGR5)), cholinergic receptor muscarinic 2 and 3 (CHRM2, CHRM3), sphingosine-1-phosphate receptor 2 (S1PR2) and MAS-related GPR family member X4 (MRGPRX4) [64].

## 4. Effects of Gut-Microbiota-Derived Metabolites on the Immune Cells

Gut-microbiota-derived SCFAs, Trp and BA metabolites exert a critical role in maintaining gut and systemic homeostasis through inhibiting inflammatory immune cells and promoting the differentiation and function of immunosuppressive cells (Figure 2)

### 4.1. Tolerogenic and Inflammatory Macrophages

Macrophages (Macs) are heterogeneous. Their phenotypes and functions can be regulated by the surrounding microenvironments. These cells are generally divided into two kinds, inflammatory (i) and tolerogenic (t, immunosuppressive) macrophages (tMacs). IMacs are involved in inflammatory immune responses, whereas tMacs suppress inflammation and retain homeostasis by producing a large amount of IL-10 and TGF-β. In the resting intestine, mature resident (tolerogenic) ly6c^low/-^CX3CR1^hi^MHCII^hi^ Macs from inflammatory Ly6c^high^ monocytes/Macs reside either within the lamina propria (LP) or the muscle layer to maintain intestinal homeostasis [65]. LP Macs can be further subdivided into mucosal and submucosal Macs [66]. The intestinal epithelium and vasculature in the intestines are lined by Mucosal Macs [67,68]. Gut-microbiota-derived metabolites such as SCFAs, Trp and BA metabolites can promote the differentiation from iMacs to tMacs (Figure 3).

**SCFAs.** SCFAs such as acetate (C2), propionate (C3) and butyrate (C4) exert an important role in maintaining immune homeostasis. Lipopolysaccharide (LPS)-mediated proinflammatory cytokines such as IL-6 could be inhibited by SCFAs. SCFAs could significantly reduce the histone deacetylase (HDAC) mRNA expression in monocytes and Macs [69]. SCFAs, especially butyrate, also negatively regulate the inflammatory signaling pathway mediated by NLRP3 (NOD-like receptor thermal protein domain associated protein 3) to inhibit the activation of Macs [70]. In addition, butyrate but not acetate or propionate can reprogram Mac metabolism toward oxidative phosphorylation to lead to an anti-inflammatory tolerogenic phenotype [71].

**Trp metabolites.** Trp metabolites (Trps) have an important role in the differentiation and function of Macs through the receptor AhR [72]. AhR-deficient mice were more sensitive to LPS-induced lethal shock [73] and produced higher amounts of tumor necrosis factor (TNF)α, interleukin (IL)-6 and IL-12. AhR was also required for *Streptococcus-* and *Salmonella typhimurium*-caused immunopathology in LPS tolerant mice [74]. In vitro studies showed that Trps-mediated suppression of inflammatory responses occurred through suppressing histamine production in the macrophages [75]. Through inhibiting LPS-induced SP1 (specificity protein 1) phosphorylation in macrophages, the AhR-SP1 complex represses histidine decarboxylase expression [75]. SP1 can bind to GC box elements (5′-GGGCGG-3′) in the promoter region [76] and is particularly important to TATA-less genes involved in the immune response [77]. It has also been found that AhR down-regulation in human disease is related to an abnormal interaction between SP1 and the AhR promoter [78]. The activation of AhR also results in a mitigated inflammatory response by LPS through a Ras-related protein Rac1 (ras-related C3 botulinum toxin substrate 1) ubiquitination-dependent mechanism, which can attenuate AKT (protein kinase B) signaling [79]. In addition, the Kyn downstream metabolite 3-HAA inhibits the LPS-mediated PI3K (phosphatidylinositol 3 kinase)/Akt (protein kinase B)/mTOR (mammalian target of rapamycin) and NF-κB (nuclear factor κ gene binding) signaling pathways and decreases the production of pro-inflammatory cytokines in the macrophages [80]. The Trp metabolite receptor AhR can also inhibit the proliferation of myeloid precursor cells [81], drive DC differentiation over Macs [72] and suppress human CD34^+^ hematopoietic precursor cells to differentiate into monocytes and Langerhans cells [82].

**BA metabolites.** BA metabolites (BAs) are essential to maintain a tolerogenic phenotype of Macs via the BA receptor TGR5 (GPBAR1) [83,84,85]. TGR5 can inhibit the release of cytokines from Macs after exposure to LPS. LPS-induced inflammation in the liver could be accelerated in TGR5-deficient mice, whereas the suppressive effects of TGR5 agonist on inflammatory cytokines could be abolished [86]. TGR5 can also block NLRP3 inflammasome-dependent inflammation [87,88]. Indeed, the TGR5 ligands and secondary BAs DCA and LCA can function as endogenous inhibitors of NLRP3 activation by activating TGR5-cAMP (adenosine monophosphate)-PKA (protein kinase A)-dependent ubiquitination of NLRP3 [87,88]. The elevated intracellular cAMP levels can induce the phosphorylation and the ubiquitination of NLRP3 to block NLRP3-dependent inflammation and NLRP3-related metabolic disorders. TGR5 activation also promotes macrophage polarization to tolerogenic-phenotype Macs [89]. The hierarchy is LCA > DCA > CDCA > UDCA > CA for TGR5 activation [90]. In addition, FXR is also essential to maintain a tolegeronic phenotype of Macs as demonstrated in FXR knockout mice [83], and it is an important negative regulator of NLRP3 by directly interacting with NLRP3 and caspase-1 [91]. FXR is recruited to iNOS (nitric oxide synthase) and IL-1β promoters and stabilizes nuclear receptor corepressor 1 (NCOR1) complexes on the promoters of these genes [92]. Several pro-inflammatory genes such as iNOS, TNFα and IL-1β are marked by NCoR1 in promoter regions, which are linked to an NF-κB responsive element. FXR also activates SOCS3 (suppressor of cytokine signaling 3), CYP450 (Cytochrome P450) and FGF19 (fibroblast growth factor 19) to inhibit inflammation and SHP (Src homology-2 containing protein tyrosine phosphatase) to inhibit NF-κB, AP-1 (activator protein-1) and NLRP3 [93,94,95,96]. PXR, a nuclear receptor, also binds to LCA [90]. PXR activation can decrease the expression of IL6, TNFα and IL8 [97].

### 4.2. Tolerogenic Dendritic Cells

Dendritic cells (DCs) link the innate and adaptive immune responses. DCs are divided into monocyte DCs (moDCs), plasmacytoid DCs (pDCs) and conventional DCs (cDCs). The cDCs can be further divided into two subsets, cDC1 and cDC2. DCs are the most efficient antigen-presenting cells and are necessary for the effective activation of naïve T cells. However, DCs can also acquire tolerogenic functions such as conventional CD11c^+^ DCs expressing perforin (perf-DCs) and CD103^+^ DCs, which participate in the central and peripheral tolerance and the resolution of immune responses. Although DCs play distinct roles in shaping T cell development, differentiation and function, tolerogenic DCs (tDCs) mainly contribute to Treg differentiation and homeostasis (Figure 3).

**SCFAs.** The SCFAs butyrate and propionate inhibit the activation of bone-marrow-derived DCs (BMDC) via suppressing the LPS-mediated expression of co-stimulatory molecules such as CD40 and the production of cytokines such as IL-6 and IL-12p40 [98]. Through modulating DCs, the SCFA butyrate also suppresses colonic inflammation and carcinogenesis [99].

**Trp metabolites.** Trp-metabolite-mediated AhR activation induces tDCs. These tDCs can limit T cell effective responses and promote the generation of Tregs. This may be because of NF-κB activation controlled by AhR, such as NF-κB expression and NF-κB/AhR protein interactions [100]. However, the molecular mechanisms involved are mostly unknown. Notably, AhR activation can indeed boost DCs to foster FoxP3^+^ Treg differentiation.

**BA metabolites.** The secondary BA DCA suppresses the LPS-induced expression of pro-inflammatory genes such as IL-6 in DCs [101]. TGR5-deficient mice could recover LPS-induced expression of pro-inflammatory genes. TGR5 activation was found to induce the differentiation of human monocytes into IL-12 and TNF-α hypo-producing DCs [102]. Studies showed that BA receptor TGR5-mediated inhibition occurred through the repression of NF-κB by TGR5–cAMP–PKA (protein kinase A) signaling [101]. In addition, the secondary BA derivative isoDCA can also limit FXR activity in DCs and confer upon them an anti-inflammatory phenotype [52]. INT-747/obeticholic acid, which could activate FXR [83], greatly attenuated the differentiation of CD14^+^ monocytes into mature DCs [103]. A reduced number of activated DCs in the colon of mice administered with INT-747/obeticholic acid was also observed. The activation of the BA receptor VDR also inhibited the production of inflammatory cytokines and the maturation of DCs [104].

### 4.3. Regulatory T Cells

Regulatory T cells (Tregs) play key roles in maintaining immune homeostasis. The differentiation and function of Tregs can be regulated by gut-microbiota-derived metabolites such as SCFAs, Trps and BAs (Figure 4). Tregs express transcription factor forkhead box protein 3 (Foxp3) and differentiate in the thymus or the periphery. These cells are the main obstacles in successful immunotherapy and active vaccination. However, other T regulatory cells such as Foxp3 negative interleukin (IL)-10 producing type 1 regulatory T cells (Tr1 cells) also play an important role in homeostasis.

**SCFAs.** SCFAs can regulate T cell function through G-protein coupled receptor (GPR) [105,106], are crucial in maintaining intestinal epithelium physiology and have a direct role in inducing Tregs in the gut. They can promote the naïve T cells toward Tregs [107]. Since SCFAs can be transported into the circulation, SCFAs also have wider systemic effects. Indeed, increased Foxp3^+^Tregs can be observed in mice provided with SCFAs [108]. The main mechanisms for SCFA-mediated Tregs include the G-protein coupled receptors (GPCRs) GPR41, GPR43 and GPR109A on the target cell surface mediating signaling and the inhibition of histone deacetylases (HDACs) to regulate gene expression [109]. The inhibition of HDAC activity can enhance gene transcription by increasing histone acetylation. Butyrate also upregulates histone H3 acetylation of Foxp3 to promote the differentiation of Tregs [108]. In addition, SCFAs such as butyrate can also condition mouse and human DCs to promote the differentiation of Tregs. After exposure to butyrate, DCs facilitate Foxp3^+^Treg differentiation and inhibit interferon (IFN)-*γ*-producing cells through indoleamine 2,3-dioxygenase 1 (IDO1) and aldehyde dehydrogenase 1A2 (Aldh1A2) [110]. Notably, SCFAs also promote the production of IL-10 in Th1, Th17 and Treg cells [111].

**Trp metabolites.** Indole and its derivatives from Trp can regulate the differentiation and function of Tregs [112,113]. The transcription factor Foxp3’s expression in Tregs can be promoted, whereas RORγ (retineic-acid-receptor-related orphan nuclear receptor gamma) in Th17 cells is inhibited by Trp metabolites. The AhR ligands 2,3,7,8-Tetrachlorodibenzo-p-dioxin (TCDD), 2-(1′H-indole-3′-carbonyl)-thiazole-4-carboxylic acid methyl ester (ITE) and 4-n-nonylphenol are linked not only to differentiation but also to the functions of Tregs in mice and humans [114,115,116,117]. The AhR activated with ITE could suppress IBD [118] and improve encephalomyelitis (EAE) symptoms [119]. Notably, *AhR* in the Tregs of spleen and lymph nodes is lower than that in the intestinal Tregs [120]. In addition, Kyn in the gut microbiota could promote differentiation of Tregs [74,121,122,123]. Mechanically, Kyn metabolites work through direct transactivation and epigenetic modifications to support Treg differentiation [123,124,125]. Indeed, 3-HAA promotes the generation of Foxp3^+^Treg cells via nuclear coactivator 7 (NCOA7) [126].

The Trp metabolite receptor AhR also promotes the development of Tr1 cells [127]. During Tr1 cell differentiation, AhR is physically associated with c-Maf to activate IL-10 and IL-21 promoters to promote the differentiation of Tr1 cells [128]. AhR activation also promotes hypoxia inducible factor-1 (HIF1)-α degradation and takes control of Tr1 cell metabolism [127]. In addition, AhR can initiate the differentiation of mucosal-homing Tim3^+^Lag3^+^Tr1 cells [129].

**BA metabolites.** BA metabolites (BAs) modulate the differentiation and function of Tregs [130]. The bile acid derivatives isoalloLCA and 3-oxoLCA can promote the differentiation of Tregs. Mechanically, these derivatives promote the generation of mitochondrial reactive oxygen species (mitoROS) [131]. Indeed, for their energy production, Tregs mainly rely on oxidative phosphorylation (oxPhos) after exposure to BA derivatives. The mitochondrial activities also promote Treg generation [132]. Nuclear receptor subfamily 4, group A, member 1 (NR4A1) is also required for the isoalloLCA-induced Treg cells [133]. IsoalloLCA can increase the binding of NR4A1 at the Foxp3 locus to enhance the expression of the Foxp3 gene [134].

The composition of the gut BA pool also modulates the colonic Tregs expressing RORγt [135]. Decreased RORγt^+^Tregs could be observed in the colon while BA metabolic pathways were genetically abolished in individual gut symbionts, whereas rescuing the intestinal BA pool increased colonic RORγ^+^Treg cells and meanwhile also ameliorated the host susceptibility to colitis. Notably, the stability of the lineage-determining transcription factors RORγ and Foxp3 in Th17 and Treg cells can be regulated by post-transcriptional modifications. In addition, the BA receptor VDR’s activation promotes the induction of Tregs [136] and reduces Th17 cell production [137]. IsoDCA also induces the generation of Foxp3^+^Tregs after reducing DC stimulatory properties by ablating FXR in DCs [52].

### 4.4. T Helper 17 Cells

Differentiation of T helper (Th) 17 cells from naïve T cells is related to professional antigen-presenting cells (APCs) and cytokines including IL-6, IL-21 and TGFβ. However, the differentiation of these cells is also affected by gut microbiota metabolites. These Th17 cells produce interleukin 17A (IL-17A), interleukin 17F (IL-17F), interleukin 21 (IL-21) and interleukin 22 (IL-22) [138].

**SCFAs.** SCFAs are crucial factors of the mucosal immune responses [139]. The gut microbiota can influence the differentiation of Tregs and Th17 cells [140]. The disequilibrium of SCFAs from the gut microbiota can damage the balance of Treg/Th17 [132]. The SCFA butyrate also decreased the proliferation and reduced the cytokine production of Th1, Th17 and Th22 cells [141]. The peroxisome proliferator-activated receptor gamma (PPARγ) and reprogrammed energy metabolism are involved in SCFA-mediated function in these cells [142].

**Trp metabolites.** Trp metabolites suppress Th1 and Th17 [143]. AhR of Trp metabolites plays a key role in Th17 cell differentiation. Indeed, IAA can decrease Th17 cells through activating AhR, downregulating RORγt and STAT3 (signal transducer and activator of transcription 3) [144]. However, studies also showed that 6-formylindolo(3,2-b) carbazole (FICZ), a Trp product, could promote T cells into Th17 cells [145].

**BAs metabolites.** Th17 and Treg cell differentiation can be controlled by BA metabolites (BAs) [131]. 3-oxoLCA and isoalloLCA can reduce Th17 cell differentiation and increased Tregs in mice[131]. Th17 cell differentiation can be inhibited by 3-oxoLCA through blocking the function of RORγt [131,146] and directly binding to RORγt [131]. Similar to 3-oxoLCA, isoLCA also suppressed Th17 cell differentiation by inhibiting RORγt [47]. RORγt is selectively expressed by Th17 and innate lymphoid cell group 3 (ILC3). It is a critical for these cells’ differentiation in chronic inflammation and autoimmune diseases [147]. Indeed, RORγt inhibition not only reduces the frequencies of Th17 cells but also provides therapeutic benefits in intestinal inflammation [148].

### 4.5. CD4^+^Th1 and Th2 Cells

CD4^+^Th1 cells are mainly responsible for cell-mediated immunity and produce interferon (IFN)-γ, IL-2 and TNF-α, whereas Th2 cells are involved in antibody production and produce IL-4, IL-5, IL-10 and IL-13 cytokines. Although T-bet and GATA binding protein 3 (GATA3) are master transcription factors for the differentiation of Th1 and Th2 cells, respectively, their differentiation and heterogeneity are usually determined by combinatorial transcription factors.

**SCFAs.** DCs from mice treated with the SCFA propionate have an impaired ability to initiate Th2 cells [149]. These DCs have a reduced expression of CD40, programmed cell death ligand 2 (PD-L2) and CD86. Notably, SCFAs can promote the microbiota’s antigen-specific IL-10 production in Th1 cells through GPR43. Mechanistically, SCFAs upregulate transcription factor B lymphocyte-induced maturation protein 1 (Blimp-1). However, SCFAs also have the potential to induce inflammatory responses [150]. SCFAs can induce Th1 and Th17 cells upon exposure to immunological challenges. A high concentration of butyrate also induces Th1 transcription factor T-bet expression.

**Trp metabolites.** Many patients with cancer often show decreased plasma Trp levels in parallel with an elevated Th1 type immune activation marker. Oral Trp supplementation suppresses antigen-specific Th1 responses at subtoxic concentrations [143]. Through IDO1-mediated Trp catabolism, synovial fibroblasts can also selectively suppress Th1 cell responses [151].

**BA metabolites.** Upon exposure to BAs, CD4^+^ T cells can maintain gut homeostasis [152]. Pols et al. revealed that unconjugated LCA inhibited the activation of primary human and mouse CD4^+^ Th1 cells to reduce TNFα and INFγ production through a BA receptor VDR-dependent mechanism [153]. A shift from Th1 to Th2 cells could be promoted by BA receptor VDR activation through c-Maf and GATA-3 [154]. A decreased number of liver-infiltrating CD4^+^ Th1 cells is associated with a good response of patients with primary biliary cholangitis to UDCA treatment.

In addition, PXR activation also inhibits T cell proliferation in both mouse and human T cells in vitro. However, CXCR5^+^CD4^+^ T follicular helper cells could be induced by BA metabolism to cause neuromyelitis optica spectrum disorder [155].

### 4.6. Regulatory B Cells

Regulatory B (Breg) cells are immunosuppressive cells that support immunological tolerance. Breg cells have multiple subsets, including immature and mature B cells, which can express IL-10, IL-35 and/or TGF-β and surface molecules such as CD9, CD1d, CD21, CD23, CD24, CD5, CD138, TIM (T cell immunoglobulin and mucin domain-1) and/or PD-L1/L2. In addition, other Breg cell subsets have also been reported such as CD1d^high^CD21^high^CD23^+^IgM^high^IgD^−^ T2 MZ (marginal zone) precursor B cells, CD1d^high^CD5^+^ CD1d^high^ CD21^high^CD23IgM^high^IgD^−^MZ B cells and CD25^+^CD69^+^ CD72^high^CD185^−^CD196^+^IgM^+^IgD^+^B cells. These cells suppress immunopathology through the production of IL-10, IL-35 and TGF-β cytokines.

**SCFAs.** Rosser and colleagues recently showed that butyrate could divert Trp metabolism toward the serotonin pathway and the production of 5-hydroxyindole-3-acetic acid (5-HIAA) [156]. 5-HIAA activates AhR in Bregs, mediating the suppressive effect in a rheumatoid arthritis model in vivo [156]. The administration of SCFAs also improved rheumatoid arthritis (RA) symptoms and increased the Breg frequency [157].

**Trp metabolites.** B cell differentiation, maturation and activation can be regulated by the Trp metabolite receptor AhR [158,159]. AhR activation regulates the differentiation and function of IL-10-producing CD19^+^CD21^high^CD24^high^Bregs [160]. AhR-deficient mice develop exacerbated arthritis with significant reductions in IL-10-producing Bregs. Our study showed that in the presence of LPS, IAA by gut microbiota could activate the transcription factors PXR, CAR and NF-κB to induce the generation of IL-35^+^ Breg cells [161]. Others also found that LPS increased the expression of p35 and Ebi3 in B cells isolated from mice [162]. The transcription factor NF-κB promoted influenza A virus (IAV)-mediated IL-35 [163].

### 4.7. B Cells

B cells play a key role in the responses to microbial infections and pathogen clearance. These B cells not only produce antibodies but also release a broad variety of cytokines. BA-metabolite-mediated VDR activation reduces the ongoing proliferation of B lymphocytes [164], induces activated B cell apoptosis [165] and inhibits Ig production [166]. However, SCFAs can also stimulate glycolysis in B cells via mTOR activation. SCFA-derived acetyl-CoA is crucial for plasma cell differentiation and antibody production [167]. SCFAs also can promote the secretion of IgA by B cells [168]. The activation of Trp metabolism is related to flavivirus-mediating B cell differentiation into antibody-secreting cells in humans [169].

### 4.8. Myeloid-Derived Suppressor Cells

Myeloid-derived suppressor cells (MDSCs) are most commonly immunosuppressive cells during chronic inflammation, especially late-stage cancers. These cells consist of two large groups of cells termed granulocytic or polymorphonuclear (PMN)-MDSCs and monocytic (M)-MDSCs. In humans, the total MDSCs are characterized by HLA-DR^low/neg^Lin^low/neg^CD33^pos^CD11b^pos^. PMN-MDSCs are identified with negative CD14 or positive CD15, whereas M-MDSCs are identified with positive CD14 or negative CD15 [170]. They use different mechanisms for immunosuppression. PMN-MDSCs mainly suppress T cell responses by producing ROS (reactive oxygen species), whereas M-MDSCs produce high amounts of NO (nitrogen oxide), Arg-1 and immunosuppressive cytokines such as IL-10, which suppress both antigen-specific and non-specific T cell responses [171]. M-MDSCs have higher suppressive activity than G-MDSCs. Taurodeoxycholate (TDCA) can increase the number of PMN-MDSCs in the spleen of septic mice [172].

### 4.9. Innate Lymphoid Cells

There are three different groups of innate lymphoid cells (ILCs), namely, ILC1s, ILC2s and ILC3s, but only ILC3s are IL-22 producers [173]. IL-22 is crucial for the maintenance of intestinal epithelial cells (IECs) and the defense against pathogens [174]. It belongs to an IL-10 family cytokine [175]. The gut microbiota has profound effects on the differentiation and functions of ILCs.

**Trp metabolites.** Trp metabolites play a critical role in the development of ILC3s. AhR activation is essential for IL-22 production in ILC3s through AhR ligands from the microbiota [176,177]. Trp metabolites are involved in mucosal immunity through AhR modulation. An impaired AhR activity in AhR knockout mice was related to reducing ILC3 and aggravating inflammatory diseases [23]. The disruption of gut-microbiota-related Trp metabolism results in reduced IL-22 in the intestinal tract, whereas the activation of AhR in ILC3 promotes IL-22 production, thereby modulating the intestinal immune response and protecting the function of the intestinal barrier. AhR also plays an important role in the differentiation of ILC3s [178,179]. Especially in the early stage after birth, AhR ligands are required for the differentiation of IL-22-producing ILC3s [180]. Mechanically, AhR not only participates in Runx3- and RORγt-mediated ILC3 development [181] but also mediates Notch and c-Kit expression [178,182]. Notably, reduced AhR signaling can cause alterations between ILC3 and ILC1 cellular populations. In addition, *AhR can also* cause IL-22 expression in the Th17 cells [183].

### 4.10. CD8^+^ T Cells

Naïve CD8^+^ T cells can produce a large number of effector cells to fight infections or tumors after antigen stimulation.

**SCFAs.** The SCFAs butyrate and propionate regulate CD8^+^ T cell activation via inhibiting IL-12 production in DCs. However, microbiota-derived SCFAs can boost CD8^+^ T cell functions by modifying the cellular metabolism [184]. The anti-tumor functions of cytotoxic T lymphocytes (CTLs) and chimeric antigen receptor (CAR) T cells can be significantly enhanced by pentanoate and butyrate [185]. Through regulating mTOR activity and cellular metabolism, acetate also promotes IFN-γ production in CD8^+^ T cells.

**Trp metabolites.** Kyn can upregulate the expression of PD-1 in CD8^+^T cells through interacting with the ligand-activated AhR [186], which mediates immunosuppressive responses. 3-HAA from the Kyn pathway causes immune suppression by inducing apoptosis in T cells through glutathione depletion [187]. However, Trp metabolites can promote CD8^+^T cells to induce apoptosis of co-cultured cancer cells, increase cancer-infiltrating CD8^+^T cells and suppress tumor growth of lung cancer in mice [188].

**BA metabolites.** BA metabolites can disrupt intracellular calcium homeostasis, which is essential for NFAT (nuclear factor of activated T cells) signaling and T cell activation [189]. 24-Norursodeoxycholic acid (NorUDCA) changes immunometabolism in CD8^+^ T cells and alleviates hepatic inflammation [190]. It has strong immunomodulatory efficacy in CD8^+^T cells, which affect lymphoblastogenesis, expansion, glycolysis and target of rapamycin complex 1 (mTORC1) signaling. BA receptor VDR activation also reduces the ongoing proliferation of T lymphocytes [164].

### 4.11. Natural Killer Cells

Natural killer (NK) cells, as a first line of defense against cancer, are powerful effectors of innate immunity. These cells can express an array of receptors to eliminate tumor cells. Kyn metabolites, particularly Kyn itself, can suppress the activity of NK cells [191] and cause cell death via a ROS pathway in NK cells [192]. These Kyn metabolites can prevent the cytokine-mediated upregulation of the specific triggering receptors responsible for NK-cell-mediated killing [193].

### 4.12. NKT Cells

NKT cells, an unusual population of T cells, can recognize lipids presented by CD1d. Gut-microbiome-mediated BA metabolism regulates liver cancer via NKT cells [194]. CXCL16 expression of liver sinusoidal endothelial cells regulated by BA can control the accumulation of NKT cells [194]. The activation of the BA receptor FXR can result in a profound inhibition to produce a potent pro-inflammatory mediator osteopontin in NKT cells [195].

### 4.13. Neutrophils

Neutrophils play a critical role in the host defense against infection. SCFA-mediated activation of GPR43 can induce the neutrophils to inflammatory sites and enhance their phagocytosis [196]. However, pro-inflammatory cytokine production such as TNFα in neutrophils can be inhibited by SCFAs [197]. SCFAs also affect neutrophil-mediated anti-HIV responses [198]. Serum BAs in liver cirrhosis promote neutrophil dysfunction [199]. Sphingosine-1-phosphate receptor (S1PR) can reduce neutrophil aggregation [200]. In addition, the Trp metabolite indole suppresses neutrophil myeloperoxidase to diminish bystander tissue damage [201].

### 4.14. CD4^+^CD8αα^+^ Cells

The intestinal epithelium contains a unique population of CD4^+^CD8αα^+^ T cells [26]. These CD4^+^CD8αα^+^ T cells can promote gut tolerance to dietary antigens [127]. They can be found in the intestine of mice colonized with *L. reuteri*. Through Trp-metabolite-mediated AhR activation, *L. reuteri* can reprogram CD4^+^ T cells into CD4^+^CD8αα^+^ cells in the gut [26]. CD4^+^CD8αα^+^ IELs can resist apoptosis and upregulate IL-15 and IL-10 in a colitis model [202].

## 5. Gut-Microbiota-Derived Metabolites and Immune-Associated Disorders

Gut-microbiota-derived SCFAs, Trp and BA metabolites have been widely related with intestinal and extra-intestinal disorders such as inflammatory bowel diseases (IBDs), chronic liver diseases, metabolic syndrome, diabetes and cancer [203,204,205,206]. Gut-microbiota-derived metabolites play a key role in inflammatory bowel disease (IBD) [204]. Metabolite disturbances including BAs and short-chain fatty acids (SCFAs) have been reported in patients with IBDs [204]. Ursodeoxycholic acid reduces the severity of intestinal inflammation in a DSS-induced mouse model of colitis [207]. Longitudinal analyses also demonstrated that certain metabolites such as tryptophan metabolites were decreased in coeliac disease [208]. The indole metabolites are dysregulated in patients with active IBD and in mouse models of colitis, and the restoration of depleted indoles reduces disease severity [209]. The metabolites from the gut microbiota can modulate the development and progression of non-alcoholic fatty liver disease (NAFLD) [210]. Tryptophan-derived microbial metabolites activate the aryl hydrocarbon receptor in tumor-associated macrophages to suppress anti-tumor immunity [211]. Interventional studies with certain bacterial strains such as *Akkermansia muciniphila* have shown effects on obesity-related parameters [206]. The tryptophan-derived metabolite IAA induces the generation of IL-35^+^B cells through PXR and TLR4 to inhibit obesity in mice [161]. Thus, the manipulation of the gut microbiota may impact the immune system and improve immune-mediated disorders. An increasing number of studies has reported the use of fecal microbiota transplantation (FMT) for the treatment of diseases such as metabolic syndrome, diabetes, multiple sclerosis, psoriasis, Crohn’s disease, cancer and Parkinson’s disease [212,213]. Typically, the modulation of the gut microbiota with the FMT method has successfully cured patients with refractory immune-checkpoint-inhibitor-associated colitis [214]. 

Notably, the composition of the gut microbiota in immunosuppressed patients such as allogeneic hematopoietic-cell transplantation is changed, which is characterized by a loss of diversity and domination by single taxa [215,216]. However, a large body of evidence has also shown that the importance of the intestinal microbiota in immunosuppressed patients. Fecal microbiota transplantation (FMT) in immunocompromised cohorts can provide protection against bacterial translocation via the introduction of a diverse microbiome and restoration of epithelial defenses [217]. Promoting microbial diversity via FMT is also likely to enhance natural barrier defenses, including anti-microbial peptides, tight junction assembly/integrity, mucus production and epithelial proliferation [217]. In addition, exclusive enteral nutrition may also cultivate the presence of beneficial microbiota and improve BA metabolism, possibly influencing disease and immune activity [218]. Several nutritional therapies have been designed not only to treat the nutritional deficiencies seen in children with active Crohn’s disease (CD) but also to correct dysbiosis and reduce intestinal inflammation [219]. Multi-donor FMT with an anti-inflammatory diet effectively induced deep remission in mild–moderate ulcerative colitis [220].

## 6. Conclusions and Perspectives

The gut microbiota harbors trillions of microorganisms in the human digestive system. These microorganisms affect the gut and systemic immunity via their metabolites such as SCFAs, Trp and BA metabolites to maintain gut and systemic homeostasis. The alteration of the gut microbiota/metabolites can lead to the onset of many diseases ranging from gastrointestinal and metabolic conditions to neuropsychiatric diseases and cancers. The effects of gut microbiota metabolites on different immune cells have important consequences not only in the onset and development of diseases but also in the diagnosis and therapy of these diseases and predictions of clinical outcomes, prognosis and immunotherapy responses such as cancer immune checkpoint blockade. With the rapid development of recent techniques, more bacterial strains to produce the metabolites (including SCFAs, Trp and BA metabolites) remain to be identified. This will be beneficial for understanding different diseases and designing targeted strategies to control the production of the metabolites for the therapy of these diseases. However, several critical techniques need to be overcome to find more gut-microbiota-derived metabolites that are potentially related to diseases.

(1)Discovery of new culture method(s) for gut microbiota. A key question for gut microbiota metabolites is whether gut microorganisms can be successfully cultured in vitro. The discovery of any new culture technique will be beneficial to the identification of gut microbiota metabolites.(2)Improvement of the metabolite analyses. For currently targeted metabolomics, the restricted standard samples have limited application, whereas for untargeted metabolomics, it is easy to produce “false positive” data.(3)Synthesis of gut microbiota metabolites. Some metabolites from the gut microbiota need to be synthesized for their functions and application.(4)Determination of immune cell subset function. With the development of single-cell sequencing techniques, more immune cell subpopulations related to the gut microbiota or metabolites will be identified. However, the functional potential of these immune cell subsets remains to be determined.(5)Establishment of new animal models. Some gut microbiota metabolites may exert their function through new mechanism(s), including receptor, signal pathway, genetic and epigenetic modification and metabolism. All of these need new animal models to explain how the metabolites exert their effects on the immune cells and/or diseases.

## Figures and Tables

**Figure 1 cells-12-00793-f001:**
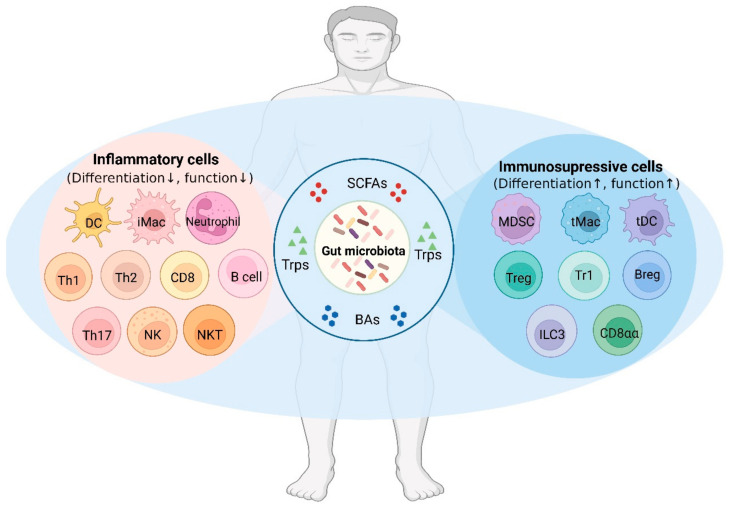
The gut microbiota maintains the homeostasis of the gut and systemic immune system through the metabolites. Metabolites from the gut microbiota such as short-chain fatty acids (SCFAs), tryptophan metabolites (Trps), and bile acid metabolites (BAs) promote the differentiation and function of immune-suppressive cells and inhibit the inflammatory cells. DC, dendritic cell; iMac, inflammatory macrophage; Th1, T helper 1; Th2, T helper 2; Th17, T helper 17; NK, natural killer cell; NKT, natural killer T cell; MDSC, myeloid-derived suppressor cell; tMac, tolerogenic macrophage; tDC, tolerogenic dendritic cell; Treg, regulatory T cells; Tr1, type 1 regulatory T cells; Breg, regulatory B cell; ILC3, innate lymphoid cell 3; CD8αα, CD4^+^CD8αα^+^ intestinal intraepithelial lymphocyte.

**Figure 2 cells-12-00793-f002:**
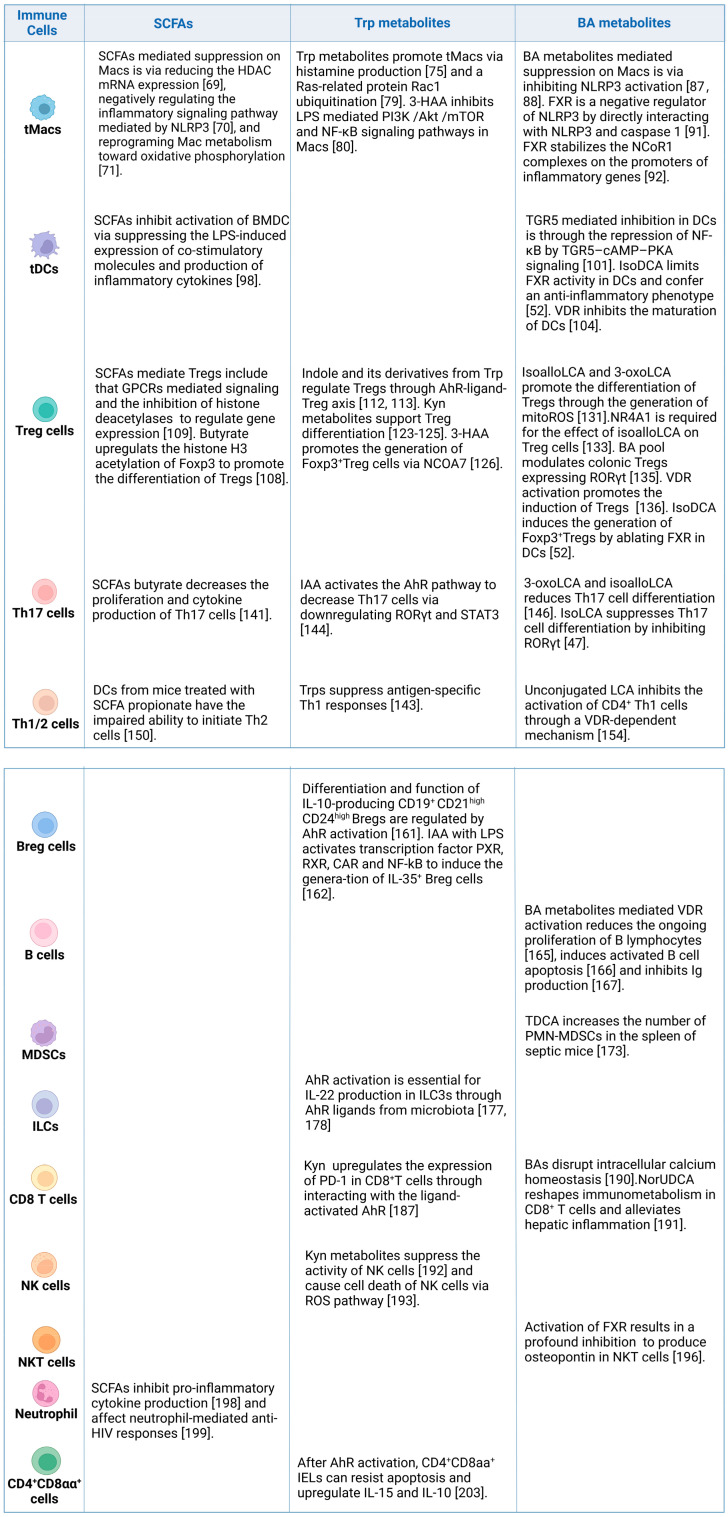
Regulation of gut-microbiota-derived metabolites in different immune cells. Gut-microbiota-derived metabolites such as SCFAs, Trp and BA metabolites can promote differentiation and function of immune-suppressive cells (such as tolerogenic macrophages (tMacs), tolerogenic dendritic cells (tDCs), myeloid-derived suppressor cells (MDSCs), T regulatory Foxp3^+^ cells (Treg), type 1 regulatory T cells (Tr1), B regulatory cells (Breg), innate lymphoid cells (ILCs) and CD4^+^CD8^+^αα cells), and inhibit inflammatory cells (such as CD4^+^T helper (Th1), CD4^+^Th2, CD4^+^Th17, CD8, B cells, natural killer (NK) cells, NKT cells and neutrophils).

**Figure 3 cells-12-00793-f003:**
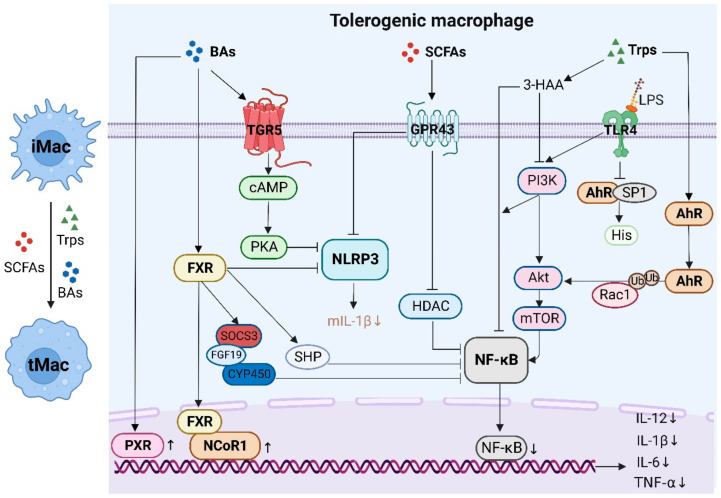
Gut-microbiota-derived metabolites promote the differentiation and function of tolerogenic macrophages through the receptors expressed in the macrophages such as short-chain fatty acids (SCFAs) through membrane receptors such as GPR43, tryptophan metabolites (Trps) through the AhR nuclear receptor and bile acid metabolites (BAs) through the TGR5 membrane receptor and/or FXR nuclear receptor. iMac, inflammatory macrophages; tMacs, tolerogenic macrophages; TGR5, Takeda G protein-coupled receptor 5; FXR, farnesoid X receptor; PXR, pregnane X receptor; NCOR1, nuclear receptor corepressor 1; cAMP, adenosine monophosphate; PKA, protein kinase A; SOCS3, suppressor of cytokine signaling 3; CYP450, cytochrome P450; FGF19, fibroblast growth factor 19; NLRP3, NOD-like receptor thermal protein domain associated protein 3; GPR43, G-protein coupled receptor 43; HDAC, histone deacetylase; NF-κB, nuclear factor-kappa B; PI3K, phosphatidylinositol 3 kinase; Akt, protein kinase B; mTOR, mammalian target of rapamycin; TLR4, Toll-like receptor 4; 3-HAA, 3-hydroxyanthranilic acid; SP1, specificity protein 1; His, histamine; AhR, aryl hydrocarbon receptor; Rac1, ras-related C3 botulinum toxin substrate 1; mIL-1β, mature interleukin -1β; TNFα, tumor necrosis factor α.

**Figure 4 cells-12-00793-f004:**
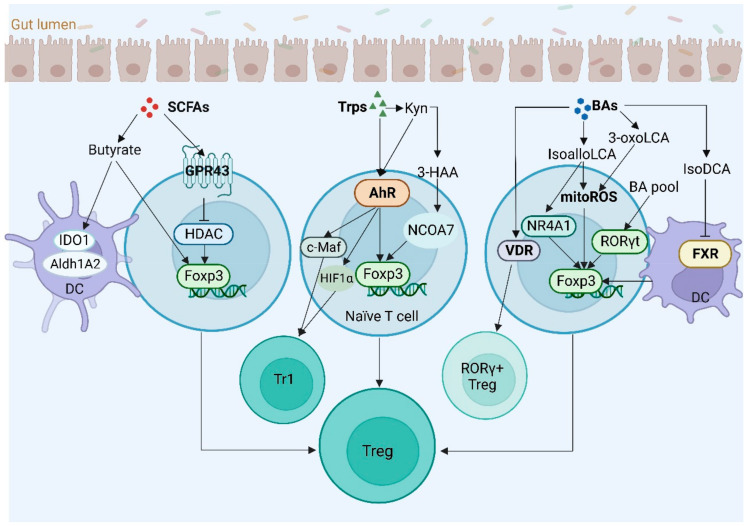
Gut-microbiota-derived metabolites promote differentiation of Treg, Tr1 and RORγt^+^ Treg cells. SCFAs, short-chain fatty acids; Trps, tryptophan metabolites; BAs, bile acid metabolites; GPR43, G-protein coupled receptor 43; HDAC, histone deacetylase; Foxp3, forkhead box protein p3; IDO, indoleamine 2,3-dioxygenase 1; Aldh1A2, aldehyde dehydrogenase 1A2; AhR, aryl hydrocarbon receptor; 3-HAA, 3-hydroxyanthranilic acid; Kyn, kynurenine; mitoROS, mitochondrial reactive oxygen species; NR4A1, nuclear receptor subfamily 4, group A, member 1; RORγt, retinoid-related orphan receptor-γt; VDR, vitamin D receptor; FXR, farnesoid X receptor; LCA, lithocholic acid; DCA, deoxycholic acid; DC, dendritic cells; Tr1, type 1 regulatory T cells.

**Table 1 cells-12-00793-t001:** Gut microbiota species and short-chain fatty acids.

SCFAs	Biosynthesis	Bacterial Species	References
Acetate (C2)	Via acetyl-CoA pathway Via Wood–Ljungdahl pathway	*Akkermansia muciniphila, Bacteroides* spp., *Bifidobacterium* spp., *Prevotella* spp., *Ruminococcus* spp *Blautia hydrogenotrophica, Clostridium* spp., *Streptococcus* spp.	[8,12,13]
Popionate (C3)	From succinate pathway From acrylate pathway From propanediol pathway	*Bacteroides* spp., *Phascolarctobacterium succinatutens, Dialister* spp., *Veillonella* spp., *Roseburia* spp., *Firmicutes, Roseburia inulinivorans, Ruminucocus* spp., *Clostridium* spp., *Eubacterium* spp., *Coprococcus* spp., and *Akkermansia muciniphila*, *Megasphaera elsdenii, Coprococcus catus, Clostridiales bacterium. Coproccus catus and Clostridium* spp. *Salmonella* spp., *Roseburia inulinivorans, Ruminococcus obeum, Eubacterium halli*	[8,14,15]
Butyrate (C4)	From butyryl-CoA acetate Co-A transferase pathway From butyrate kinase pathwayFrom lactate and acetate	*Anaerostipes* spp., *Coprococcus catus, Eubacterium rectale, Eubacterium hallii, Faecalibacterium prausnitzii, Roseburia* spp., *Roseburia intestinalis, Roseburia insulinivorans, Clostridiales bacterium, Anaerostripes* spp, *Coprococcu* spp., *Costridium symbiosum* and *Faecalibacterium prasnitzii*. *Coprococcus comes* and *Coprococcus eutactus*. *Eubacterium hallii* and *Anaerostipes* spp	[8,14,16,17]

**Table 2 cells-12-00793-t002:** Gut microbiota species and tryptophan metabolites.

Metabolite	Biosynthesis	Bacterial Species	References
Indole	Form Trp metabolism by tryptophanase	*Clostridium limosum, Bacteroides ovatus, Enterococcus faecalis* and *Escheichia coli*	[20]
IAA	From Trp metabolism through the oxidative and reductive pathways by tryptophan 2-monooxygenase or acyl-CoA dehydrogenase	*Clostridium sporogenes Clostridium bartlettii* and *Bifidobacterium* spp.	[22,23,24]
IPA	From Trp metabolism through the oxidative and reductive pathways by tryptophan 2-monooxygenase or acyl-CoA dehydrogenase and via phenyllactate dehydratae and acyl-CoA dehydrogenase	*Clostridium sporogenes Clostridium bartlettii and Bifidobacterium* spp. and *Peptostreptococcus spp*	[22,23,24,25]
IA	From Trp metabolism via phenyllactate dehydratae and acyl-CoA dehydrogenase	*Peptostreptococcus* spp.	[25]
Skatole	From Trp metabolism by decarboxylation of IAA	*Bacteroides* spp. and *Clostridium* spp.	[24,26]
IA1d	From Trp metabolism via an aromatic amino acid aminotransferase (ArAT) and indolelactic acid dehydrogenase (ILDH)	*Lactobacillus johnsonii, L. reuteri, L. acidophilus* and *L. murinus*	[27]
Tryptamine	From Trp metabolism via a Trp decarboxylase enzyme	*Ruminococcus gnavus and Clostridium sporogenes.*	[28]
3-hydroxyanthranilic acid (3-HAA)	From Trp metabolism via eukaryotic Kyn pathway	*Pseudomonas, Burkholderia, Stenotrophomonas, Xanthomonas, Shewanella*, and *Bacillus*	[29]

**Table 3 cells-12-00793-t003:** Gut microbiota species and bile acid metabolites.

Bile Acids (BAs)	Biosynthesis	Bacterial Species	References
Conjugated BAs	From primary BAs to conjugate with other amino acids	*Clostridium bolteae*, *Bacteriodetes Bacteroides vulgatus, Firmicutes Lactobcillus rumini*, *Actinobacteria Hungatella hathewayi*, *Bacterorides vulgatus*, *Lactobacillus ruminis*, *Holdemania filiformis*, *Clostridium scindens*	[39,40,41]
Deconjugated BAs	Via deconjugating by bile salt hydrolases (BSHs)	*Lactobacillus* spp., *Clostridium* spp., *Bifidobacterium* spp., *Enterococcus* spp., and *Bacteroides* spp.	[41,42,43,44,45,46,47,48]
Secondary BAs (DCA, LCA)	From deconjugated BAs through deconjugation, dehydroxylation, oxidation and epimerization	*Clostridium clusters* XIVa, IV, XI, *C. scindens*, *C. hylemonae* and *C. perfringens*, *Blautia producta, Eggerthella lenta*, *Clostridium scindens*.	[49,50]
3-oxoLCA and isoLCA	Convert LCA to 3-oxoLCA and isoLCA	*Adlercreutzia, Bifidobacterium, Enterocloster, Clostridium, Collinsella, Eggerthella, Gordonibacter, Monoglobus, Peptoniphilus, Phocea, Raoultibacter*, and *Mediterraneibacte*	[35]
Ursodeoxycholic acid (UDCA)	Conversion of 7-oxo-LCA	*Clostridium absonum, Stenotrophomonas maltophilia, Ruminococcus gnavus* and *Collinsella aerofaciens*	[51,52]
UDCA	Conversion of *7*α-*epimerization*	*Clostridium baratii*	[31]

## Data Availability

All data generated or analyzed during this study are included in this published article.

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
