# Peer review of "Gut-Microbiota-Derived Metabolites Maintain Gut and Systemic Immune Homeostasis"

_cells, 2023, doi:10.3390/cells12050793_

Round 1

Reviewer 1 Report

Authors well-summarized the effects of gut microbiota derived metabolites (SCFAs, Trp and BAs) on the main immune cells including macrophages, DCs, T and B cells. Generally, these metabolites promote the differentiation and function of immunosuppressive cells and inhibit the inflammatory cells. But in some cases, they promote the inflammation, thereby together maintaining the systemic immune homeostasis. Overall, it is a good, updated review summarizing the relationship between gut metabolites and different immune cells and worth to be published.

Few points need to be addressed:

  1. Metabolites from gut microbiota include vitamins and several amino acids, why the authors specifically focus on short chain fatty acids (SCFAs), tryptophan (Trp) and bile acid (BA)? A sentence could be included if they are the 3 major metabolites.

  1. Authors listed tons of microbiota in the first part, but not well-organized. In general, what are the dominant gut microbiota, bacteria, fungi or viruses, and what are the most common strains of them respectively? If there is any preference for the strains to generate a specific metabolite?

  1. There are several different types of immune cells included in the 4th part, such as macrophages, DCs, T and B cells, but less is described for neutrophils. If there is a particular reason, like neutrophil expression is low in gut or so?

  1. Many typos need to be fixed, especially 3.1. Font size should be consistent within a paragraph.

Reviewer 2 Report

I confirm the need for a digression regarding the importance of the intestinal microbiota in immunosuppressed patients. This is also due to the extensive digression present in the paper on the role played by the numerous effector cells of immunity, which are obviously deficient in patients undergoing chemotherapy.
In particular, it would be useful to carry out an examination of the data present in literature on the microbiota of patients undergoing hematopoietic stem cell transplantation, on the role of enteral nutrition in the preservation of the host's microbiota and in the data present in this setting in "stool" transplantation. Regarding the sentence below I would also suggest: Preparation of genetically modified microorganism. It is necessary to generate genetically modified microorganisms for the function and potential clinical application of gut microbiota metabolites. However, it is not easy to overexpress and/or knock out the genes in microorganism, especially overexpression of gene(s), which needs to look for highly efficient promoter(s). 1) what types of genes are we referring to? 2) which methods of "recognition" of these genes have been used (sanger, NGS, what type of NGS) and which are the genetic and/or promoter nucleotide sequences, etc. considered useful for the purpose. 3) analyze the literature on this topic, generate a table with the references and insert them in the bibliography. In my opinion, this conclusion must be supported or eliminated.

Reviewer 3 Report

This reviewer manuscript by Wang et al discussed three classes of microbiota metabolites,  short-chain fatty acids (SCFA), tryptophan (Trp) and bile acids (BA), specifically concerning their interactions with various cell types important in the maintenance of immune homeostasis. A large number of references were cited to support the discussion.  Authors may wish to revise the current version of the review addressing several specific issues:

1).  Contents in page 3 to page 5 are essentially a duplicate of what had been stated in Tables 1-3. The contents should be revised and drastically reduced to avoid redundancy.

2). The current version looked like a stockpile of references related to different metabolites on different cell types, all in a parallel format lack of integration and in-depth discussion of systemic effects, as the title so implied. An effort should be made to condense the review but make better reference toward changes in microbiota diversity, quantity of metabolites, immune functional balances, and their association with specific diseases or pathological conditions.

3) Authors could consider to discuss a smaller number of cell types but to make better discussions about each chosen cell type. For example, the discussion about MDSC in page 14 could only be regarded as descriptive at the most. If authors choose to discuss MDSCs, then the discussion should be in much more detail separating G-MDSCs from M-MDSCs as they could have completely different functional implications.

4) English editing the entire review manuscript is needed.
